# *Physcomitrium patens*: A Single Model to Study Oriented Cell Divisions in 1D to 3D Patterning

**DOI:** 10.3390/ijms22052626

**Published:** 2021-03-05

**Authors:** Jeroen de Keijzer, Alejandra Freire Rios, Viola Willemsen

**Affiliations:** 1Laboratory of Cell Biology, Department of Plant Sciences, Wageningen University & Research, 6708 PB Wageningen, The Netherlands; Jeroen.deKeijzer@wur.nl (J.d.K.); alejandra.freirerios@wur.nl (A.F.R.); 2Laboratory of Molecular Biology, Department of Plant Sciences, Wageningen University & Research, 6708 PB Wageningen, The Netherlands

**Keywords:** asymmetric cell division, proliferative cell division, division plane positioning, *Physcomitrium*, gametophore initiation

## Abstract

Development in multicellular organisms relies on cell proliferation and specialization. In plants, both these processes critically depend on the spatial organization of cells within a tissue. Owing to an absence of significant cellular migration, the relative position of plant cells is virtually made permanent at the moment of division. Therefore, in numerous plant developmental contexts, the (divergent) developmental trajectories of daughter cells are dependent on division plane positioning in the parental cell. Prior to and throughout division, specific cellular processes inform, establish and execute division plane control. For studying these facets of division plane control, the moss *Physcomitrium* (*Physcomitrella*) *patens* has emerged as a suitable model system. Developmental progression in this organism starts out simple and transitions towards a body plan with a three-dimensional structure. The transition is accompanied by a series of divisions where cell fate transitions and division plane positioning go hand in hand. These divisions are experimentally highly tractable and accessible. In this review, we will highlight recently uncovered mechanisms, including polarity protein complexes and cytoskeletal structures, and transcriptional regulators, that are required for 1D to 3D body plan formation.

## 1. Introduction

Proper development of three-dimensional multicellular organisms requires accurate relative positioning of cells and acquisition of new cellular identities. These processes drive the formation of tissues with specialized functions that allow organisms with higher complexity. The placement of new cells follows a robust plan/blueprint in order to achieve anatomies specific for each species. In plants, cellular migration is extremely limited because cells are encased in their cell walls. Therefore, plant development relies on controlled directional cell expansion and changes in the orientation of their division plane. For this, cells must have accurate spatial information prior to initiating formative divisions, including cues with respect to organ/organismal axes and cellular polarization. Disordered formative divisions early in development can be lethal, and later ones can seriously affect the formation of important organs, affecting survival and reproduction [1,2,3,4,5].

With the importance of accurate spatial information to position the division plane during asymmetric divisions in plants, uniquely tailored and fascinating cellular mechanisms that generate and interpret this information have come to light. Conceptually, three phases can be distinguished where such mechanisms are active (summarized in Figure 1). Firstly, before division, information that can break symmetry along a particular cellular axis within the parental cell must be established. Both cell-intrinsic factors and extrinsic factors supply this information (Figure 1A). Cell-intrinsic factors are required for internal symmetry breaking and often function via cortically located polarity protein complexes. Cell-extrinsic factors include cues from surrounding tissue that are mostly biochemical but also mechanical in nature. In plants, the continued exposure of daughter cells to extrinsic positional information is typically also involved in driving further cell fate divergence after division has taken place [6]. Cell-intrinsic and extrinsic factors are not independent and can operate synergistically or antagonistically in providing positional information [7].

Secondly, just prior to mitotic onset, the symmetry-breaking cues are translated into processes that anticipate the selected division plane. These chiefly include nuclear positioning and the establishment of structures that help specify the orientation of the incipient division apparatus (Figure 1B). A prominent structure involved in the latter is a specialized zone at the cell cortex that acts as a division plane landmark (termed the “cortical division zone”; see glossary). The cytoskeleton plays important roles in the formation of such pre-mitotic structures involved in division plane specification. Finally, as cell division is executed, the position and orientation of the division apparatus is actively controlled (Figure 1C). The new dividing wall is formed in the final stages of division by the phragmoplast (see glossary). Its communication with the previously specified cortical domain then fine-tunes the final orientation of the dividing wall separating the two daughter cells (Figure 1C).

In this review, we focus on recent research covering several major mechanisms and molecular players that function during these three phases to control plant asymmetric cell divisions. This will be complemented by a brief discussion of the key transcriptional and hormonal regulators that trigger and govern asymmetric cell division. We will focus specifically on how various developmental steps in the bryophyte *Physcomitrium patens (P. patens)* can collectively provide a well-suited platform to aid in the study of asymmetric cell division. *P. patens* also benefits from a host of molecular genetic tools and a simple body plan with mostly monolayered tissues and organs that are easily accessible by microscopy.

### Glossary

**Formative cell division:** cell division that generates daughters with different identities; also called formative asymmetric cell division (ACD).**Proliferative cell division:** cell division that generates daughters of the same identity; also called **symmetric** cell division (**SCD**).**Cell fate/cell identity:** commitment to cell type-specific genetic programs.**Cell division plane:** Actual or forecast plane physically separating two daughter cells.**Symmetry-breaking/cellular polarization:** unequal distribution of molecules and cellular components. Required for important processes like differential cell fate acquisition of two daughter cells.**Cortical Division Zone (CDZ):** A membrane and cell wall-associated domain at the cell cortex established at or just before mitotic entry that specifies a plane in the parental cell through which daughter cells will ultimately be partitioned (see also Figure 1B). The CDZ has a dynamic composition that includes cytoskeletal and membrane-bound components, and functions as landmark for the correct insertion of the nascent dividing wall constructed by the phragmoplast.**Pre-prophase band (PPB):** Ring-shaped assembly of the microtubule cytoskeleton and associated proteins that transiently appears before the onset of cell division. The overall orientation of the PPB appears to be inherited from that of the interphase cortical microtubules, and its position correlates with that of the CDZ.**Phragmoplast:** Plant-specific cellular apparatus that brings about physical separation of two newly formed daughter cells (cytokinesis) at the end of cell division. It consists of two opposing sets of microtubules, in the center of which, small, membranous building blocks are assembled into a radially expanding precursor of the new dividing wall. Insertion of this precursor at the parental wall occurs at the site specified by the CDZ.

## 2. Developmental Stages of *P. patens* Are Marked by Characteristic Cell Divisions and Establishment of New Growth Axes

At the start of its lifecycle (i.e., after a spore germinates), a moss plant establishes itself by outgrowth of filamentous tissue, called protonemata (Figure 2A). These filaments consist of two types: a slow-growing, photosynthetically active type and a rapidly expanding type with underdeveloped chloroplasts, called chloronemata and caulonemata, respectively. Both types expand exclusively by highly polarized tip growth to effectively explore the plant’s immediate environment [8]. Initially, primary filaments have a chloronemal identity, which, after several division rounds of the tip cell, can transition to a caulonemal identity. Notably, the division planes in chloronemata are perpendicular to the growth axis, while those in caulonemata are consistently slanted (Figure 2(B1)). The physiological or developmental relevance of the slanted cross walls for the organism has not yet been established. The chloronema-to-caulonema identity transition is controlled by the plant hormone auxin and a set of conserved transcription factors [9,10]. Interestingly, auxin signaling is important for division plane positioning in other plant systems [11,12], hinting that similar roles may be encountered in moss. Overall, the simple patterning and unidirectional expansion of these filamentous tissues allows us to address fundamental questions regarding developmental decisions taking place at the (sub)cellular level, such as polarity formation and division plane control.

Further developmental progression in the protonemal filament relies on branching. Here, a new growth axis is established on a pre-existing filament, allowing the tissue to expand in a second dimension (Figure 2(B2); recently reviewed in [13]). Branching is initiated by a subapical cell and is under the control of hormonal and carbon-related signaling [10,14,15], although it also shows probabilistic elements, with a variable frequency of branching occurring in a typical filament. Branching normally occurs on the apex-directed side of a mother cell and is oriented according to environmental inputs like gravity and light [16,17]. Prior to visible outgrowth of a new branch, the mother cell undergoes intracellular reorganization (cell polarization) to bring its nucleus and cell division machinery towards the designated branching site. Recent work has established that cell polarization is relayed through Rho of plants (ROP) proteins, and that nuclear guidance by actin and microtubules plays a major role during the branching process (see Section 3.1) [18,19]. The formed outgrowth will be separated from the subapical mother cell at the moment of cell division and continues to grow at its tip as a secondary protonemal apical cell.

During the juvenile protonemal stage, development of so-called buds is initiated. These buds give rise to leafy shoots with three-dimensional tissue growth, on top of which the gamete-forming organs are ultimately formed; hence their name, gametophores (Figure 2A). Buds are initiated on older subapical caulonemal cells by the formation of a bulge similar to that during a branching event. Contrary to branching, though, after an initial transition division, the bulge will instead swell in a diffuse manner and then divide in an oblique manner. This oblique division will generate an apical–basal and medial–lateral axis [20] (Figure 2(B3)). Subsequent divisions initiate three-dimensional development proper and are precisely positioned to give rise to a tetrahedral apical cell (further discussed below), which principally drives further gametophore development [20,21]. Thus, a series of asymmetric divisions accomplishes the transition to the 3D body patterning of the more mature gametophore tissues from a precursor tissue with a 2D growth mode. Despite the similarities between branch formation and bud initiation on a protonemal parental cell, the morphology of the outgrowth and the angle of the division plane distinguish the two [20,22]. The switch to the gametophore developmental program involves several distinct layers of transcriptional and hormonal regulation (indicated by yellow nuclei in Figure 2B; reviewed in [23] and further discussed in Section 6 and Section 7). However, the precise moment the competency of a subapical caulonemal cell to produce buds is determined is unclear, but fate determination seems at least to be initiated in the parental cell before the division leading to this transition takes place [22]. 

The tetrahedral cell at the apex of a gametophore fulfils a meristematic function to drive shoot growth and development. This tetrahedral cell has four sides, along three of which, consecutive cutting faces produce daughter cells (Figure 2(B4)). The overall division plane orientations of the consecutive divisions rotate slightly left- or right-handed with respect to the main axis of the stem [24]. The positioning of these divisions planes is likely very precise, as it underpins the phyllotactic pattern of the gametophore (reviewed in [25]).

## 3. Signaling Molecules Driving Cell Polarization in Moss

Several protein families and protein domains involved in cell polarity are conserved across kingdoms, suggesting the presence of conserved underlying molecular mechanisms. Here, we discuss some of these major protein families and how they act as intrinsic cellular cues for cell polarity in land plants.

### 3.1. ROPs

Small GTPases (Rho, Rac and CDC42) are highly conserved in yeast and animals. Small Rho-GTPases are known as master regulators of cell polarity in eukaryotes. They serve as a positioning cue, having effects in several subcellular processes like re-arrangement of cytoskeleton elements and exocytosis (reviewed in [26]). 

In land plants, a family of small GTPases (Rho of Plants (ROPs)) is also present; however, they are sometimes referred to as RACs. Like other small GTPases, ROP/RACs have a GTPase catalytic domain that allows them to transition from a GDP-bound inactive to a GTP-bound active state. Active ROPs cluster membrane domains to which they can attach via lipid modifications of their *C*-terminus. Upstream of ROPs, ROP guanidine exchange factors (ROP-GEFs) promote their activation at the membrane. Active ROPs can regulate different effector proteins, ultimately controlling subcellular events involved in many different biological processes (pathogen responses, hormone responses, cell growth, etc.; reviewed in [26]). One important family of plant-specific ROP effectors is named RICs (ROP-Interacting CRIB-containing proteins). RIC proteins are characterized by containing the ROP-interactive CRIB (Cdc42-and Rac-Interactive Binding) motif, which is able to physically interact with GTP-bound ROPs. Different functions have been assigned to few members of the *Arabidopsis* RIC family that involve cytoskeleton reorganization (either actin or microtubule filaments) [27,28]. Based on sequence homology, there is only one putative RIC protein in *P. patens* [29]. Sequences encoding for RIC proteins were not found in the genomes of other members of the bryophyte clade, hornworts and liverworts (Freire-Rios, unpublished).

While ROPs have been shown to have a mechanistic role for some specific plant formative divisions (e.g., stomata formation in monocots [30]), it is not clear if they are necessary for cell fate specification, as was demonstrated in animals [31]. The fact that this has never been successfully shown in plants could be due to the high number of ROP family members and their redundant functions in model flowering plants. *P. patens*, with only four almost identical ROP protein family members, has been proposed as a model to study the role of ROPs and their effectors. Recently, it has been suggested that accumulation of tagged PpROP4 not only predicts the sites of filamentous outgrowth (tips and branches), but also the position of new division planes in protonema filaments during cell divisions [18,19]. Because PpROP4 protein accumulation precedes and is partly responsible for nuclear movement towards the division site during filament branching [18,19], it has been speculated that PpROP4s act on microtubule organization. However, the underlying molecular mechanisms of these responses remain unknown. In an ongoing study specifically focused on asymmetric cell divisions in *P. patens*, it was observed that deletion of one of the PpROP members leads to plants hampered in establishing filaments with a caulonema identity, and abnormally shaped gametophores. Interestingly, these defects are rescued by the deletion of the single putative PpRIC effector (Freire-Rios, unpublished). More broadly in bryophytes, a study in *Marchantia polymorfa* (a liverwort) showed a role for ROP signaling in plant development [32]. The downstream mechanism, though, has not yet been described.

### 3.2. SOSEKIs

The SOSEKI family of polarly localized proteins has been recently identified and is conserved in land plants ([33,34]. Originally identified in *Arabidopsis*, each of the five family members in this species was found to accumulate in a different corner of the cell, from which the name SOSEKI (Japanese for “cornerstone”) was derived [34]. SOSEKIs represent an outstanding and intriguing class of polar proteins because their accumulation appears to be independent of the conventional cellular trafficking pathways involved in polar protein delivery [34]. Functionally dissecting the constituent protein domains revealed that SOSEKIs associate with the plasma membrane at specific cell edges via a centrally located domain, where they oligomerize via their *N*-terminal domain. Collectively, these properties lead to their highly polarized accumulation at cell corners [33,34]. The *N*-terminal oligomerization domain is functionally equivalent to the DIX domains found in the animal kingdom. In animals, DIX domains are broadly involved in forming a polarized scaffold during planar polarity signaling also via oligomerization. The oligomerization functions to effectively recruit downstream signal transduction components, despite low individual protein–protein affinities between DIX-containing proteins and such components. Analogous SOSEKI-associated signaling components in plants are beginning to be discovered. In *Arabidopsis*, ANGUSTIFOLIA (AN) has been identified as a factor recruited by SOSEKIs. Since *AN* mutants show polarity-related phenotypes, like rounded cells and misorientation of cell division planes [35,36], its SOSEKI association could be an important aspect of its functioning.

An ancestral role of SOSEKIs within the land plants is strongly suggested by the highly similar behavior of labeled SOSEKIs in the bryophytes *Marchantia polymorpha* and *P. patens* [33]. The SOSEKI family of *P. patens* (PpSOKs) consists of 11 members and, so far, the subcellular localization of four members has been analyzed. Two out of these four PpSOKs show polar subcellular localization. Notably, one of the analyzed members, PpSOK2, is absent during filamentous growth but is specifically observed during early 3D bud and gametophore leaf development. This suggests some form of specificity in the functionality of SOSEKI protein family members in *P. patens*. Further study of such members with a clear correlation to certain developmental steps could help elucidate their role in 3D moss development and establish the degree of functional conservation of SOSEKIs more broadly.

## 4. Peptide-Mediated Intercellular Signaling during Moss Development

Extrinsic cues and their perception mechanisms are important to relay information on the tissue context and surroundings to an asymmetrically dividing cell. Furthermore, the transmission of molecules can play an important role in the establishment or maintenance of cellular identities after the division has taken place [37]. It has been shown, for example, that in flowering plants, intercellular movement of transcription factors and microRNAs plays a role in cell identity acquisition and maintenance (reviewed in [38]), but in mosses, this remains a widely unexplored field and will not be covered in this review. So far, signaling through small peptides remains the most studied form of intercellular communication in mosses.

An evolutionary conserved module involved in the spatial coordination of stem cell proliferation consists of small secreted chains of amino acids (peptides) and their cognate membrane-bound receptor proteins. The peptides belong to the CLAVATA3 (CLV3)/EMBRYO SURROUNDING REGION-related (CLE) family and are ligands for plasma membrane-located leucine-rich repeat receptor-like kinases (LRR-RLKs). Upon CLE perception, the LRR-RLKs typically modulate the proliferative activity of a stem cell population, although some CLE–LRR-RLK combinations function in other developmental and physiological processes as well (reviewed by [39,40]). Since the CLE peptides can freely diffuse in the apoplastic space, they often relay information between neighboring cells. This spatial aspect of CLE-mediated signaling has been found to be able to fulfil dual roles in the various meristems found in seed plants. Firstly, it generates a feedback loop between the different zones of a meristem such that stem cell homeostasis can be coordinated [39]. This was first described for the shoot apical meristem [41] and later appeared to be recurrent in root meristems as well [42]. Secondly, it was found that in vascular meristems, a CLE gradient provides instructions for division plane determination [43]. Thus, the CLE–LRR-RLK module constitutes a bona fide intercellular communication relay to orient formative cell divisions (such as that outlined in Figure 1B). Intriguingly, this function was found to also occur in bryophyte species, including *P. patens* [44,45]. In moss, loss of CLE and LRR-RLK function led to the misplacement of division planes, starting during the earliest developmental steps of gametophore initiation [45]. This suggests that instructing the orientation of the division plane during 3D development could be an ancestral function of the CLE–LRR-RLK signaling module. Which downstream components are used by the cell to implement the positional instructions conveyed by the CLE gradient are currently unknown. This prominent open question could be addressed using *P. patens* bud development as a simple and accessible model.

## 5. Role of the Cytoskeleton in Division Plane Control

The cytoskeleton is one of the key structural components to bring about cell division, and thus many regulatory mechanisms for division plane control appear to impinge on these subcellular components. In the context of asymmetric division, they are thus notable candidates to convert internal and external cues into the required division plane position/orientation. Classically, in seed plants, two microtubule configurations have been strongly connected to designate and build a new division site. The first is the microtubule array that develops from the mitotic spindle and brings about construction of the separating wall between the two nascent daughter cells: the phragmoplast (see glossary). The function of the phragmoplast is to assemble smaller building blocks supplied by the secretory system into a straight, disc-shaped precursor of the dividing wall [46]. The second is a microtubular ring in the form of the preprophase band (PPB) that forecasts the division plane prior to the start of cell division proper. While the PPB microtubule structure is transient, its position and orientation coincide with a ring-shaped domain at the plasma membrane with a specialized molecular makeup that persists throughout mitosis, called the cortical division zone (CDZ; see glossary). Although dynamic, the CDZ fulfils the role of a “molecular memory” to guide expansion of the phragmoplast such that, ultimately, the nascent wall connects to the parental wall at the CDZ-defined position. For recent comprehensive reviews on the CDZ’s molecular makeup and function, see [47,48,49,50].

A generalized causal role for PPB microtubules in defining the CDZ is currently unclear. On the one hand, the net orientation of microtubules at the cellular cortex which is preserved by the PPB correlates well with decisions about where to position the division plane [51,52,53]. Furthermore, under dynamic reorientation of cortical microtubules in response to changing cell shapes or mechanical stress patterns, the PPB maintains an indicator role for division plane positioning [54,55]. On the other hand, however, findings in the *Arabidopsis* root have established that PPB microtubules are not prominently required for division plane selection and CDZ establishment, but instead fine-tune the plane [56,57]. Consistently, important CDZ markers like POK1 (Phragmoplast Orienting Kinesin 1) localize correctly in absence of PPB microtubules, albeit less efficiently [56]. It is thus highly conceivable that PPB-independent mechanisms convey information towards “imprinting” of the CDZ and thus the division plane selection process, possibly still incorporating orientational information from earlier cortical microtubule arrays. A prime future challenge will thus be to explore such alternate mechanisms and expose the molecular pathways involved.

*Physcomitrium* has the potential to broaden our knowledge on how pre-mitotic microtubular constellations and tentative alternatives function during division plane establishment. In *P. patens*, divisions in protonemata and early gametophore development do not exhibit a PPB, whereas those in later gametophore tissues do [21,58,59]. This transition in presence of a PPB is mirrored by a pronounced role for the functionally conserved regulator of PPB formation TONNEAU1 later in the moss lifecycle, chiefly during leaf development [60]. However, precisely controlled divisions still take place in early moss development. For example, the tilting of the division plane in caulonemal cells (Figure 2(B1)) is partly affected by external cues like gravity [16], signifying that molecular pathways must be in place to establish a deviating division plane. Similarly, bud development relies on finely tuned division planes that take place in absence of PPBs [20,21]. This must thus mean that PPB-independent mechanisms are operational in these tissues.

One such pre-mitotic structure with clear links to division plane determination is a cytoplasmic cloud of microtubules typically associated with one (but occasionally more) side of the nucleus [21,59]. Using early bud development as a model of oriented cell division, chemical disruption of the cloud highlighted its role in correctly initiating the main axis of the spindle and subsequent phragmoplast [21]. The spatial information provided by this structure thus seems vectoral in nature (Figure 1C). Whether there is an interplay with any cortically located division plane determinants remains unknown, as these have not yet been identified in bud development. In the later developmental stages of moss, where PPBs are formed, the cytosolic microtubule clouds are still observed, although, here, their functional relevance is unexplored [21]. Whether the lack of a canonical PPB microtubule configuration in certain moss tissues represents an evolutionary loss (and is thus a derived state) or whether it is indicative of ancestral mechanisms that evolved earlier is currently unknown. The fact that moss presents us with a gradient of two distinct preprophase microtubule configurations that both function in establishing division plane orientation will allow us to better study and ultimately understand how diverse coupling mechanisms between pre-mitotic cytoskeletal structures and division plane specification evolved and function.

Beyond pre-mitotic events providing landmarks or setting the initial conditions for the division apparatus, continued control over its position and axis is required. The cytoskeleton and associated proteins are generally implicated in control of these parameters [50,61]. In the study of these processes, the divisions in *P. patens* protonemata and buds have proven to be experimentally accessible models. For example, recent findings show that the microtubule-associated protein TPX2 is essential for maintenance of a central spindle position along the apical–basal axis in buds [62]. This defect could surprisingly be compensated by actin cytoskeleton disruption, which, under normal conditions, does not significantly interfere with spindle/phragmoplast positioning [21,62]. These findings open new avenues for study on the mechanisms controlling “tugging” of the division apparatus during mitosis and its implications for division plane positioning. Another principle that positions the division apparatus involves its communication with the CDZ. In protonemal moss cells, a physical link between the two, mediated by actin and associated myosin Class VIII motor proteins, is established, which assists in division plane guidance [63]. Such cytoskeletal bridging between the CDZ and the division apparatus is documented in diverse other plants as well [47,64]. High-resolution imaging and in vitro reconstitution experiments are currently promising techniques to deliver the details on the distances across which bridging acts, and how and where the required forces are generated [65].

## 6. Cellular and Transcriptional Signal Transduction Mechanisms for Asymmetric Cell Divisions

The different (sub)cellular phenomena setting up polarity axes and executing asymmetrical divisions discussed above are under the control of biochemical and genetic regulation. Since asymmetrical divisions typically participate in distinct steps of plant developmental programs, they must be effectively wired into developmental signaling mechanisms. Several biochemical, transcriptional and hormonal regulators that facilitate this in land plants in general and *P. patens* in particular have come to light.

### 6.1. Defective Kernel1

A pivotal and intensively investigated protein required for setting up the correct division planes during land plant development and cell type specification is DEK1 (Defective Kernel 1). The name Defective Kernel 1 is derived from a class of maize mutants with a defective endosperm, in which the gene was first isolated [66,67]. It later appeared that the protein family founded by maize DEK1 was highly conserved across land plants. DEK1 family proteins all share an *N*-terminal region of several membrane-spanning domains and a *C*-terminal cysteine protease moiety homologous to calpain proteases. The membrane-associated domain has been implicated in responsiveness to external stimuli, while the calpain protease confers most, if not all, the biological activity required for downstream signaling [68,69,70]. Findings from various plant systems broadly connect DEK1 function to the correct formation and specification of new cell layers at the boundary of plant organs during early development (e.g., aleurone and epidermal layers) [71,72,73,74]. The embryo lethality of many DEK1 mutants in seed plants has, however, made cellular and molecular details of DEK1 functioning sparse.

Recently, further mechanistic study of DEK1-type protein function has been greatly expedited on two fronts by functional analysis of *P. patens*, DEK1 (PpDEK1). Firstly, the evolutionary trajectory of *P. patens* as representative of a basal land plant branch, enabled comparative studies within the land plant clade that clearly established the functional conservation of the calpain protease domain of PpDEK1 with its orthologs in flowering plants [69]. Secondly, because PpDEK1 specifically functions during and after the transition to 3D growth and the 2D protonemata are sufficient for laboratory manipulation, PpDEK1 null mutants are not lethal. This makes *P. patens* an attractive model organism for dissecting DEK1 molecular pathways. So far, this has been exploited to establish the roles of several subdomains in the protein and their interoperability within the protein family [75,76]. Furthermore, by combining the precise embedding of a fluorophore tag within the DEK1 protein with live-cell imaging, a highly polarized distribution of PpDEK1 to the faces of recently divided cells was discovered during bud formation [77]. These advances illustrate how the diverse body pattern transitions and molecular genetic toolkit of *P. patens* can benefit the study of key cellular processes involved in plant development. Overall, DEK1 is emerging as a transducer of critical intra- or extracellular signals to spatially coordinate formative divisions. Encouraging results have revealed that the stimulus for DEK1 could be mechanical in nature [68]. Further open challenges remain, such as the elucidation of the downstream target(s) of the calpain protease domain, and the cause and functional relevance of the subcellular polarization.

### 6.2. Transcriptional Regulation by APBs

Transcriptome analysis comparing wild-type *P. patens* with *Δdek1* revealed various putative downstream genes, including family members of the AINTEGUMENTA, PLETHORA and BABY BOOM (APB) AP2-type transcription factor family that have orthologs in *Arabidopsis*. Specifically, the expression analysis showed upregulation of *PpAPB2* and *PpAPB3* in lines where *DEK1* was deleted [76], suggesting that their expression is repressed by PpDEK1. The in total four moss PpAPBs have collectively been shown to be indispensable for the initiation of gametophores from protonema cells. Quadruple *apb* knockout plants revealed budless protonemata, and overexpressing *PpAPB4* resulted in enhanced bud formation. Translational fusion of all PpAPBs to reporter proteins showed that they are expressed in emerging gametophore cells but not in secondary apical protonema cells. Taken together, these points indicate that one of the outputs of PpDEK1 downstream signaling might be the suppression of PpAPB-mediated gametophore initiation [76,78]. Furthermore, the ubiquitin-associated protein NO GAMETOPHORES 1 (PpNOG1) was found to positively regulate the number of gametophores formed and played a role in the orientation of the division plane [79]. A recent model for the regulation of three-dimensional growth in *P. patens* proposed that PpNOG1 and PpDEK1 act antagonistically to regulate the expression of *PpAPB* genes [23].

PpAPBs orthologs in *Arabidopsis* belong to the PLETHORA/AIL (PLT) family of transcription factors. As in *P. patens*, the AtPLT family members show a high level of functional redundancy: single mutants do not show an obvious phenotype, while higher-order mutants exhibit phenotypes from ceased roots to embryo lethality. Overexpression of *PLT2* in *Arabidopsis* induces cell divisions and reorientation of cell division planes (Willemsen, unpublished results). In *Arabidopsis*, *PLTs* are regulated via an autoregulatory feedback loop with auxin and the auxin efflux facilitators called PIN proteins to maintain an auxin gradient and root meristem function [80,81]. Furthermore, it has been shown that prolonged high auxin levels generate a narrow *PLT* transcription domain in the roots’ apical meristem. From there, it forms a PLT protein gradient which is generated through cell-to-cell movement and further diluted by cell divisions. The different levels of PLT proteins in the root define the different developmental zones of the root tip (i.e., high levels preserve the stem cell niche and promote cell division, whereas low levels result in cell differentiation) [82]. This indicates that PLT function is required for stem cell initiation, stem cell maintenance and cell division, as well as positioning the orientation of the cell division plane [80,83] (Willemsen, unpublished results). *PpAPBs* are also under control of auxin, which could indicate that this is a conserved functional link [78].

## 7. Hormonal Regulation of Asymmetric Cell Divisions in Moss

### 7.1. Auxin

The hormone auxin is a key factor for regulation of plant development, and it has been suggested that this function was adopted during the evolution of early land plants. Orthologs of the auxin sensing and response machinery like the TIR1/AFB-AUX/IAA co-receptors and three classes of the ARF transcription factors of *Arabidopsis* are conserved in bryophyte genomes [84,85,86,87,88,89,90,91,92,93,94,95,96,97]. Auxin movement in *P. patens* is mediated by efflux carriers, including the membrane-localized PINs [98,99]. The *P. patens* genome has three PIN homologs (*PpPINA*, *PpPINB*, *PpPINC*), which encode for proteins that are polarly localized in the plasma membrane [98,99]. Additionally, there is one atypical PIN version (PpPIND) which resides intracellularly at what is likely the ER membrane and resembles PIN5 in *Arabidopsis* [99]. PINA proteins are polarly localized in the membrane between the protonemal cells, with the highest abundance at the tip [99]. The localization of PpPINA-C is tipward, indicating that the source is probably located at the base of the colony and that auxin is transported towards the far end of the filament [10,99] *PINA* over-expressors show enhanced auxin export and branch numbers on caulonema cells, while *pinapinb* double mutants show reduced export but are not impaired in branch formation. This might indicate that the PIN proteins have highly overlapping functions and higher-order mutants are required to induce severe effects, like in *Arabidopsis* [98,100]. The chloronema-to-caulonema transition is induced by auxin, indicating that auxin can change the identity of tip cells. This transition is also required to facilitate cells that can induce gametophore formation [9,84,94,101]. It was recently demonstrated in *Arabidopsis* that auxin can have a direct role in the establishment of plant cell polarity by promoting ROP clustering in membrane domains that could be locally activated [102,103,104]. This raises the question as to whether this could be a recurrent factor downstream of auxin during the regulation of cell division planes required for gametophore formation.

### 7.2. Cytokinin

It is known that the plant hormone cytokinin is involved in cell fate transition and bud formation, but hitherto, the underlying mechanism has been unknown. Different scenarios can be thought of, and one possibility is that the function of cytokinin is required for symmetry-breaking, as has been observed in *Arabidopsis* [105,106]. Additionally, in *Arabidopsis*, it has been shown that PLTs (via auxin) and the cytokinin response regulator (ARR12) antagonistically regulate each other to control cell size and organ growth [107]. In *P. patens*, it has been shown that *PpAPB* genes activate cytokinin biosynthetic genes, which will induce the formation of initial gametophore cells [23]. Subsequently, a proposed feedback mechanism containing PpDEK1, PpNOG1 and CLAVATA makes sure that the divisions within the initial gametophore are oriented correctly [23,79]. Additionally, NO GAMETOPHORES 2 (PpNOG2) has been identified, and knockout mutants showed a misregulation of auxin-responsive genes [108]. New players that expand our knowledge on the feedback loops guiding gametophore formation are continuously being identified, but their precise spatio-temporal activity patterns remain unresolved.

## 8. Future Directions

The mechanisms that link the described cellular processes involved in division plane positioning and the overarching gene and hormone regulatory networks are only beginning to be understood. For instance, the links among cell polarity, cytoskeleton and transcriptional regulation is obscure. An important step towards understanding them is to identify new players in specific and accessible formative divisions. For this, a simple biological model and state-of-the-art techniques are required.

Our ability to observe events at the cellular level can be difficult in multi-layered organs of big, three-dimensional organisms (e.g., model flowering plants). Mosses instead offer researchers accessibility to a multitude of cell divisions of varying complexity during their lifecycle: 1D in filament extension, 2D in filament branching and 3D in bud formation (Figure 2). All are readily present during routine moss cultivation, but can also be experimentally halted or triggered by the researcher (e.g., by changing culture conditions or by hormonal induction).

Despite differences between the early cell divisions in branch- or bud-forming cells, the underlying molecular mechanisms specifying their fates is yet unknown. The characteristics that make moss an amenable system for cell biology studies should be accompanied by rigorously timed genetic expression studies that could provide early molecular markers of cell fate. We foresee that application of novel approaches like single cell sequencing will help with the identification of cell fate maps, opening up great avenues for the study of formative cell divisions and their molecular control during one- to three-dimensional plant body patterning.

## Figures and Tables

**Figure 1 ijms-22-02626-f001:**
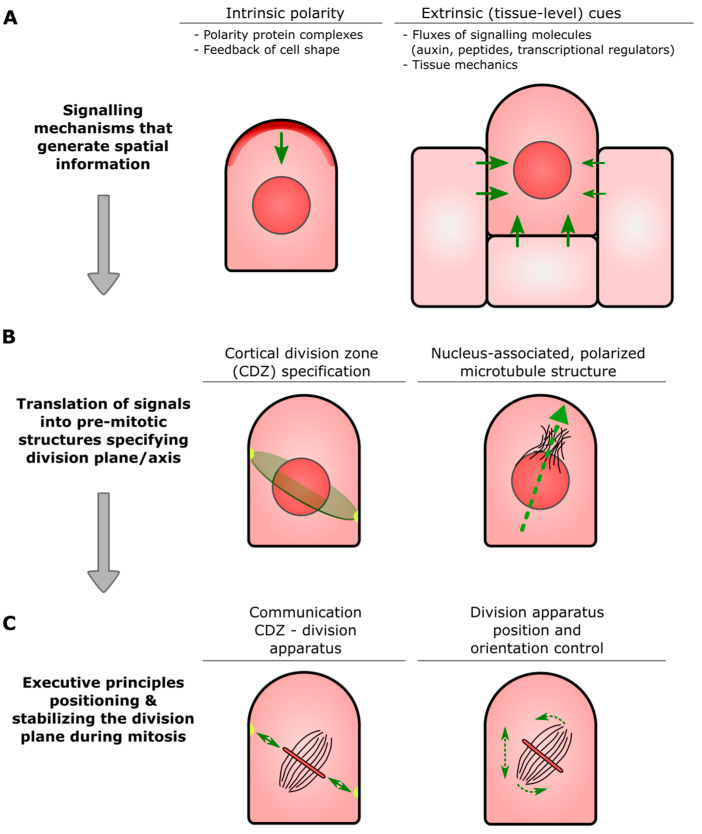
Generalized schematic representation of the processes and factors necessary for correct positioning of division planes during (asymmetrical) plant cell division. (**A**) For a parental cell to polarize, positional information is required. Positional information is generated by both cell-intrinsic factors (left) and extrinsic factors from surrounding tissues (right). Cell-intrinsic factors chiefly act at the cell cortex, where they can establish polarized signaling domains (red). Extrinsic signaling occurs via (unequal) exposure to cues, examples of which are provided. (**B**) As the parental cell is about to undergo mitosis, specific structures arise that will instruct the prospective division apparatus on the position and orientation of the desired division plane. The most prominent structure is the cortical division zone (CDZ; light green) that marks the partitioning plane (left). Additionally, polarized microtubule structures that are associated with the nucleus affect the axis along which division will take place (right). (**C**) As mitosis is executed, continuous control over the position and orientation of the division apparatus is required for proper division plane control (right). Final guidance of the division apparatus constructing the nascent dividing wall towards the CDZ is achieved by physical and/or biochemical communication between these two structures (left).

**Figure 2 ijms-22-02626-f002:**
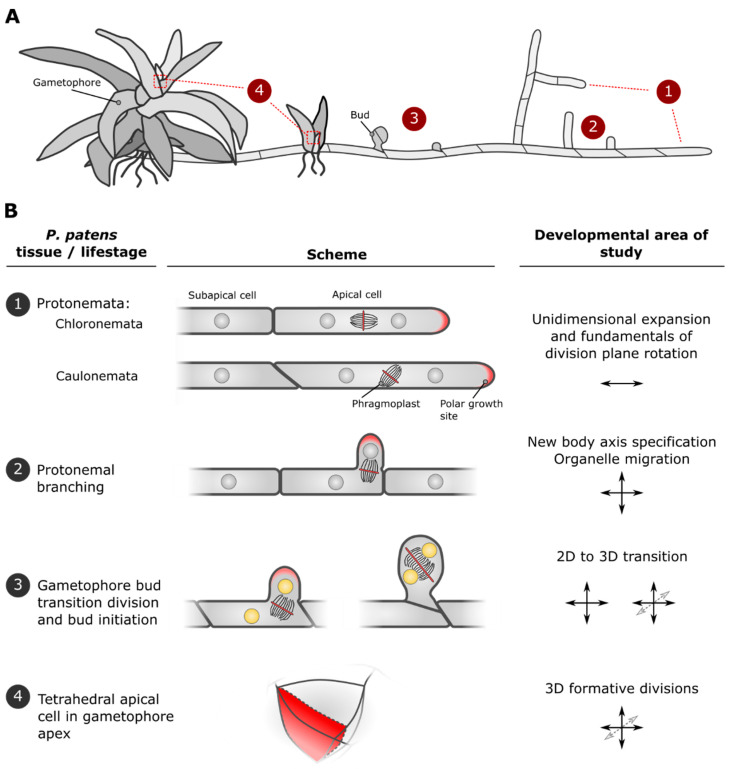
Developmental progression in *Physcomitrium patens* and the accompanying cellular phenomena that can be studied. (**A**) Schematic overview of stages in *P.*
*patens* (gametophytic) development. Cellular outlines (protonemata/buds) or tissue outlines (gametophores) are depicted. Tissue types predominantly associated with juvenile up to adult phases are arranged right to left. Numbers correspond to particular tissues and/or life stages, with oriented cell divisions leading to tissue type specification or changes in growth axes that are further detailed in B. (**B**) Four *P. patens* tissues/life stages where various aspects of cell division plane orientation and the establishment of new organismal axes can be studied: (1) Two types of filamentous protonemata (chloronemata + caulonemata) both grow by polarized, unidimensional cell expansion at their apex. The former produces division planes (red line) perpendicular to the growth axis, while the latter exhibits tilting of the division apparatus (phragmoplast), leading to slanted division planes. (2) A secondary growth axis (indicated by arrows) within protonemal tissue can be established by branching of subapical cells. This involves cell polarization and control over nuclear position and division plane orientation. (3) From the juvenile protonema, a transition to 3D developing gametophores can be initiated. This starts by outgrowth of a bud accompanied by cell divisions with specific division plane orientations that establish new organismal axes. Initiation of this developmental program is regulated by distinct transcriptional and hormonal pathways (indicated by yellow nuclei). (4) The apex of the bud ultimately gives rise to a singular stem cell with three cutting faces (one is indicated). Its continued production of daughter cells and their further developmental trajectories drive gametophore morphogenesis.

## Data Availability

The data presented in this study are available on request from the corresponding author.

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
