# Peer review of "Physcomitrium patens: A Single Model to Study Oriented Cell Divisions in 1D to 3D Patterning"

_ijms, 2021, doi:10.3390/ijms22052626_

Round 1
Reviewer 1 Report
Overall this review on Physcomitrium patens as a model to study mechanisms of cell division is well written and informative.
Specific comments:
Title: “Body plan” does not mean “growth dimension”, these are two different things, using body plan in this context is confusing
Line 82: “P. patens also benefits from a host of molecular genetic tools and a simple body 82 plan that is easy accessible by microscopy”. Can you explain what is a simple body plan?
Line 142: Branching principles have been reviewed recently in
Line 157: What is the definition of 3 dimensional leafy shoots ?
Line 211 : « has never successfully been shown” > been is missing
Author Response
Dear reviewer,
First of all, we would like to thank you carefully reading our manuscript and for your comments, which were useful and improve the manuscript.
Title: “Body plan” does not mean “growth dimension”, these are two different things, using body plan in this context is confusing
>We agree with this and have changed the title.
Line 82: “P. patens also benefits from a host of molecular genetic tools and a simple body 82 plan that is easy accessible by microscopy”. Can you explain what is a simple body plan?
> Thank you for this comment, we have included the different growth stages in Fig 2A, which shows the simplicity of the body plan and have changed the indicated sentence in the text.
Line 142: Branching principles have been reviewed recently in
> Sorry, for not including this reference. It is included in the text.
Line 157: What is the definition of 3 dimensional leafy shoots ?
>We have changed the sentence to make it more clear.
Line 211 : « has never successfully been shown” > been is missing
> Thank you for your comment, we had overlooked this and it is now added.
Reviewer 2 Report
This review is about the mechanisms that control asymmetric cell divisions in plants using the moss Physcomitrella patens as a model system. The authors are first explaining the basic principles behind establishment of formative cell divisions in plants, which involves three major steps: First, establishment of cell polarization, second positioning of the division plane and third separation of the daughter cells. The authors describe the most important protein complexes and signaling pathways including transcription factors and hormonal signals involved in these processes in P. patens. They also explain how these mechanisms are conserved in higher plant species and what is known about orthologous proteins involved. They conclude by outlining how molecular tools available in P. patens can contribute to understanding important basic developmental mechanisms in plant development.
This is a very clear and informative manuscript about key cellular processes in plant biology. The clear and well-structured text is accompanied by two well-designed figures that are nicely illustrating the key questions that are being addressed. I am highly recommending publication of this manuscript.
Author Response
Dear reviewer,
We would like to thank you for carefully reading our manuscript.